# Understanding Input Transformation-Based Attacks via Target Function Space Expansion

## Abstract

Research on transfer-based adversarial attacks provides critical insights into distinctions among Deep Neural Networks (DNNs), revealing their vulnerabilities when exposed to unseen noise. Among these transfer-based adversarial attacks, input transformation-based attacks are popular due to their simplicity and effectiveness. However, their mechanisms remain poorly understood, potentially hindering advancements in DNNs. This work explores the mechanism of the attacks, suggesting that 1) when trained with input transformations, models can improve transformation invariance by capturing diverse features from transformed inputs rather than transformation-invariant features. Therefore, given a surrogate model $f_s$ trained with input transformations $\varphi$, adversarial attacks can leverage these transformations to expand the target function space $f_s \circ \varphi$, thereby effectively and rapidly improving adversarial transferability, as domain shifts are mitigated; 2) input transformation-based attacks enhance adversarial transferability by expanding the target function space. Such transformations effectively act as modifications to the target model, thereby improving attack robustness against diverse models; and 3) L2-normalization should be incorporated into the attack paradigm to mitigate gradient imbalance during adversarial example generation. This imbalance arises from domain shift variability induced by different transformations. Based on the findings, we design a simple transformation-based attack called SimAttack. It achieves a mean attack success rate of 95.4% on 12 models, and some of the generated examples are also effective against GPT 4.1.

## 1 Introduction

To identify the deficiencies of DNNs, researchers investigate the way to deceive a model by adding perturbations to inputs, which refers to an adversarial attack. Later, it reveals that these adversarial attacks can deceive another model while crafting noisy inputs for one model (i.e, surrogate models). Thus the transferability study of adversarial attacks has come into focus and many novel transfer-based attacks are proposed to improve the transferability of adversarial attacks, such as gradient-based methods (Goodfellow et al., 2014; Kurakin et al., 2018; Dong et al., 2018; Fang et al., 2024), input transformation-based methods (Xie et al., 2019; Zou et al., 2020; Lin et al., 2024; Zhu et al., 2024a; Guo et al., 2025), model-related methods (Zhang et al., 2023; Xiaosen et al., 2023; Wang et al., 2024b), ensemble-based methods (Liu et al., 2016; Chen et al., 2023a;b) and generation-based methods (Naseer et al., 2019; Zhu et al., 2024b).

Among these transfer-based adversarial attacks, input transformation-based attacks are popular due to their simplicity and effectiveness, and the update process for such attacks can be uniformly formulated as

$$x_t^{adv} = x_{t-1}^{adv} + \alpha \cdot sign(\sum_i \nabla_{x_{t-1}^{adv}} J(f_s(\varphi_i(x_{t-1}^{adv})), y)), \tag{1}$$

where $x_t^{adv}$ denotes the adversarial example at the $t$-th iteration, and $\alpha$ is the step size. This parameter is typically set to $\epsilon/T$ (i.e., the perturbation budget $\epsilon$ divided by the total iterations $T$), ensuring the perturbation intensity remains within budget. The $f_s$ represents the surrogate model, while $\varphi_i$ denotes the $i$-th random transformation.

Here, inspired by the work Guo et al. (2025), we provide a more intuitive definition. Input transformation-based attacks leverage random transformation $\varphi(\cdot)$ to construct multiple compos-

ite functions $f_s(\varphi_i(\cdot))$ belonging to the function space $f_s \circ \varphi$, totaling $I$ instances. The input $x_{t-1}^{adv}$ is used to query these functions to obtain the updated adversarial example $x_t^{adv}$, which can enable the updated data $x_t^{adv}$ to attack all the functions in the function space. This process can enhance adversarial transferability, allowing the adversarial example $x_t^{adv}$ to effectively attack other models.

From the above definition, this transformation modifies the surrogate model, not the input data. Each transformation instance $\varphi_i(\cdot)$ can be viewed as a function $\varphi_i : R^{C \times H \times W} \to R^{C \times H \times W}$, and then all the compositions $f_s(\varphi_i(\cdot))$ can construct a function space. If the target models $f_{tar}(\cdot)$ reside in this function space, such attacks successfully transfer to them. We argue that input transformation-based attacks leveraging $\varphi(\cdot)$ can enhance model transformation invariance of adversarial examples by adopting transformations that bridge feature-capture gaps between surrogate and target models, thereby boosting adversarial transferability. The model transformation invariance refers to the fact that adversarial examples remain effective even when the target model undergoes a transformation.

However, if the surrogate model fails to extract meaningful features from these transformed data $\varphi_i(x_{t-1}^{adv})$, the resulting updates may become severely noisy. Consequently, if the transformed data $\varphi_i(x_t^{adv})$ resembles in-domain training data, adversarial attacks leveraging such transformations can generate effective perturbations. This efficacy stems from models' inherent strength in processing in-domain data. The Methodology section (i.e., Sec. 3) will investigate the effect of domain shift induced by data transformations and the feasibility of its mitigation.

Additionally, a critical concern emerges: if models learn transformation-invariant features, random transformations cannot expand the function space, thereby rendering the method ineffective. This limitation arises because capturing the same features before and after the transformation will result in identical example gradients before and after the transformation. Typically, transformations $\varphi(\cdot)$ are employed during training to enhance model transformation-invariance, defined as consistent outputs for original and transformed inputs. Intuitively, this is because models may capture transformation-invariant features. This issue will also be discussed in the Methodology section.

This study mainly investigates the aforementioned issues, with contributions summarized as follows:

- When trained with input transformations, models can improve transformation invariance by capturing diverse features from transformed inputs rather than transformation-invariant features. Consequently, given a surrogate model trained with input transformations, adversarial attacks can leverage these transformations to expand the target function space, thereby effectively and rapidly improving adversarial transferability, as models excel on in-domain data.

- Input transformation-based attacks enhance adversarial transferability by expanding the target function space. Such transformations effectively act as modifications to the target model, thereby improving attack robustness against diverse models.

- During adversarial example generation, varying domain shifts across different transformations can cause gradient imbalance and excessive noise. To address this, we introduce gradient normalization while employing a large set of random transformations.

- Based on these findings, we propose SimAttack, a simple yet effective transformation-based attack. This method leverages transformations used in surrogate model training along with other effective ones, incorporating gradient normalization to achieve state-of-the-art results in the experimental evaluations.

## 2 RELATED WORK

This work mainly focuses on the mechanism interpretability of input transformation-based attacks, but also touches on mechanisms for model-related attacks and utilizes some ideas from gradient attribution. Therefore, in this section, we introduce these three types of related work.

**Input Transformation-Based Attack.** One of the most popular approaches is the input transformation-based attack due to its effectiveness and simplicity. The input transformation-based attack elaborates transformations to enhance adversarial transferability. DIM Xie et al. (2019) randomly resizes and adds padding to input examples to improve adversarial transferability. Consequently, DEM Zou et al. (2020) calculates the average gradient of several DIM's transformed im-

ages to further improve adversarial transferability. Then, many novel transformations are presented, which calculate the average gradient of the transformed images to improve adversarial transferability. For example, SSM Long et al. (2022) randomly scales images and adds noise in the frequency domain. SIA Wang et al. (2023) splits the image into blocks and applies various transformations to each block. DeCowA Lin et al. (2024) augments input examples via an elastic deformation to obtain rich local details of the augmented inputs. L2T Zhu et al. (2024a) optimizes the input-transformation trajectory along the adversarial iteration, achieving great performance. BSR Wang et al. (2024a) randomly shuffles and rotates the image blocks to generate adversarial examples with great adversarial transferability. OPS Guo et al. (2025) observes a mirroring relationship between model generalization and adversarial example transferability and uses transformation and random perturbations to generate adversarial examples. The mechanism underlying this approach is not well understood. To mitigate this, our work demonstrates that data transformation can help adversarial attacks avoid low-transferability perturbations by guiding adversarial examples to attack more models.

**Model-Related Attack.** This approach improves adversarial transferability by changing properties of surrogate models, such as data transformation and structural changes. SGM Wu et al. (2020) utilizes more gradients from the skip connections in the residual blocks. MTA Qin et al. (2023) trains a meta-surrogate model, whose adversarial examples can maximize the loss on a single or a set of pre-trained surrogate models. AGS Wang et al. (2024b) trains surrogate models with adversary-centric contrastive learning and adversarial invariant learning. VDC Wang et al. (2024a) adds virtual dense connections for dense gradient back-propagation in attention maps and MLP blocks, without altering the forward propagation. Our work investigates the role of input transformations.

**Gradient-Based Feature Attribution.** This approach delves into identifying the importance of input features to the model's output. CAM Zhou et al. (2016) identifies discriminative regions that the model uses to make a prediction through the linearly weighted summation of activation maps from the last convolution layer. Grad-Cam Selvaraju et al. (2017) introduces a general method that uses the gradients w.r.t. the activation map to measure the channel importance. Later, some novel methods Xu et al. (2020); Zhuo & Ge (2024) are proposed to improve performance by reducing noise, but the key idea remains unchanged. Our work uses feature attribution to demonstrate whether there are differences in the features captured by the model before and after data transformation.

# 3 METHODOLOGY

## 3.1 ROLE OF TRANSFORMATION IN MODEL TRAINING

Among previous transfer-based adversarial attacks, some methods Wu et al. (2020); Qin et al. (2023); Wang et al. (2024b;a), referred to as model-related attacks, enhance adversarial transferability by strategically modifying surrogate models. This demonstrates that the properties of surrogate models play a crucial role in adversarial example generation. Meanwhile, input transformations are widely used in model training to improve transformation invariance, implying they influence surrogate model characteristics. Then **what is the influence, and what difference do they make when using in input transformation-based attacks compared to other transformations?**

Typically, data transformations are considered to provide models with transformation-invariance. In other words, whether a model processes transformed data or original data, it can produce invariant outputs. Intuitively, this may arise from models capturing transformation-invariant features through data transformations. However, the results shown in Fig. 1 demonstrate that this is not true. As shown in Fig. 1, we utilize the feature attribution to show the difference between the captured features of original and transformed data and calculate the similarity between the results, which can be formulated as:

$$Cos(\nabla_x J(f_s(\varphi(x)), y), \nabla_x J(f_s(x), y)), \tag{2}$$

where $Cos(\cdot, \cdot)$ computes cosine similarity, and other symbols align with Eq. 1.

To investigate whether models can capture transformation-invariant features via data transformation, we train models on CIFAR-100 Krizhevsky et al. (2009) transformed by random rotation and resize-padding. We then compare gradient-based feature attribution maps between transformed and non-transformed inputs using Eq. 2. To show the influence of data transformations used in model training, beyond the transformation (i.e., rotation) employed during training, we introduce a transformation (i.e., block shuffle) unused in training for comparison. If models successfully capture

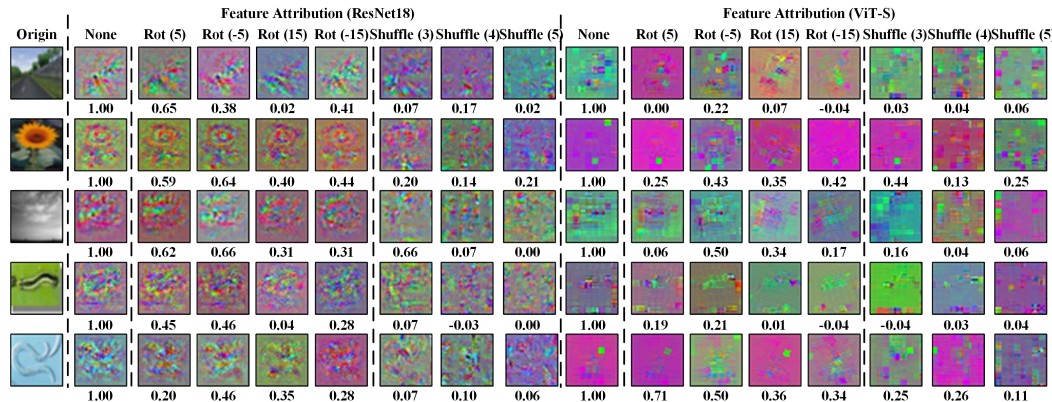

Figure 1: Similarity between gradients of inputs with and without transformation (i.e., the columns labeled "None" and others). Similarity can be calculated by Eq. 2. The used models are trained using data transformed by random rotation and resize-padding. The "None" column refers to inputs with no transformation, "Rot" to inputs with rotation, and "Shuffle" to inputs with block shuffle. Specifically, the "Rot (5)" indicates a rotation of 5 degrees, while "Shuffle (3)" indicates that inputs are split into several blocks and these blocks are randomly shuffled. The results suggest that although models can yield invariant outputs when processing transformed data, the models capture different features derived from the transformed data rather than invariant features.

transformation-invariant features, the feature attribution maps of the transformed and original data should be identical or highly similar. This can be formulated as

$$\nabla_x J(f_s(\varphi(x)), y) = \nabla_x J(f_s(x), y). \tag{3}$$

This invariance will lead to the degradation of input transformation-based attacks when utilizing transformations used in surrogate model training. The degradation can be formulated as

$$
\begin{aligned}
x_t^{adv} &= x_{t-1}^{adv} + \alpha \cdot sign(\sum_i \nabla_{x_{t-1}^{adv}} J(f_s(\varphi_i(x_{t-1}^{adv})), y)) \\
&= x_{t-1}^{adv} + \alpha \cdot sign(\nabla_{x_{t-1}^{adv}} J(f_s(x_{t-1}^{adv}), y)).
\end{aligned}
\tag{4}
$$

However, the results in Fig. 1 demonstrate that regardless of whether a transformation was used in training, models produce different responses to the transformed inputs. This suggests that models cannot capture transformation-invariant features via data transformation. Instead, the models establish more projections $\varphi(x) \rightarrow y$. This supports that input transformation-based attacks do not degenerate into non-transformation-based attacks when utilizing the transformation used in surrogate model training. Therefore, we can utilize the transformation used in surrogate model training as the random transformation $\varphi$ in the attack, guiding adversarial examples to target a set of functions $f_s(\varphi_i(\cdot))$ ( belonging to a function space) instead of a single function $f_s(\cdot)$. Straightforwardly, this can improve the adversarial transferability of the adversarial examples while having little impact on the attack success rate against the surrogate model $f_s(\cdot)$, since the transformed data $\varphi_i(x)$ remains in-domain for the surrogate model $f_s(\cdot)$. To further demonstrate these points, we train surrogate models with and without transformations and then use the models to generate adversarial examples with and without the transformations. The results of these adversarial examples are shown in Fig. 2.

The results in Fig. 2 show that surrogate models trained with transformations can leverage the transformations to generate better adversarial examples superior to those generated without the transformations. This also supports that data transformation helps models establish more projections $\varphi(x) \rightarrow y$ rather than capturing transformation-invariant features. Also, the comparison of surrogate models with and without transformations reveals that models exhibiting better generalization capabilities typically generate more effective adversarial examples.

In summary, the role of data transformation is to help models capture different features from transformed data rather than capturing transformation-invariant features. Therefore, input transformation-based attacks can leverage the transformation used in surrogate model training as the random transformation $\varphi$ in the attack, guiding adversarial examples to target a set of functions $f_s(\varphi_i(\cdot))$ to improve transferability.

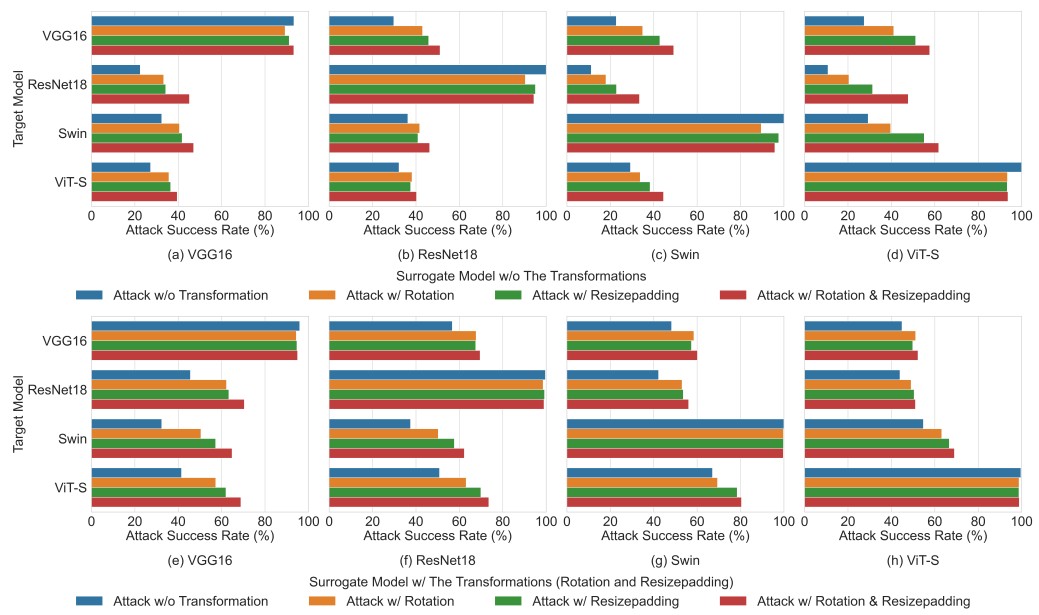

Figure 2: Attack success rate of input transformation-based attack with and without the data transformation used in surrogate model training. The surrogate models are trained on CIFAR-100, and the transformation number $I$ of the attack is 100. We evaluate the performance on the data that can be classified correctly by all the models.

## 3.2 MECHANISM OF INPUT TRANSFORMATION-BASED ATTACKS

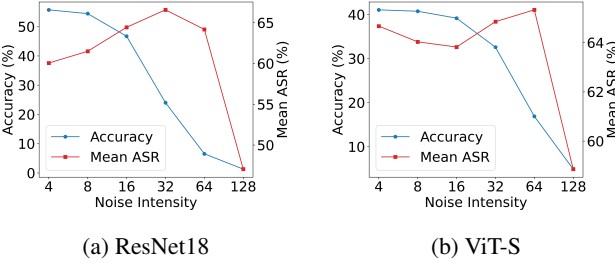

Figure 3: Accuracy of noisy inputs and mean attack success rate of adversarial attacks that employ random uniform noise as the transformation. The transformation number $I$ is 100, with noise intensity defining the bounds of uniform sampling. We run 5 times and take the mean. This illustrates the trade-off between domain shift induced by data transformations $\varphi(\cdot)$ and the range of target function space $f_s(\varphi(\cdot))$.

The results in Fig. 2 show that input transformation-based attacks introduce the transformation (i.e., rotation and block shuffle) to improve adversarial transferability, even when surrogate models are trained without them. This suggests that, beyond the transformations used in surrogate model training, the attacks can introduce additional transformations to enhance transferability. However, this introduces a trade-off: while data transformations $\varphi(\cdot)$ expand the target function space $f_s \circ \varphi$, they simultaneously induce domain shifts that degrade the model. To further illustrate this clearly, we generate adversarial examples with noise of different intensities, and the results are presented in Fig. 3. We can expand the function space $f_s(\varphi(\cdot))$ by enhancing the bound of random noise, as a larger bound encompasses smaller variations. The domain shifts are demonstrated through the accuracy of images with varying noise intensities. As shown in Fig. 3, severe domain shift leads to model degradation. However, compared to downstream tasks, its impact on adversarial attacks is significantly less pronounced. This phenomenon may arise because the attack mechanism can leverage feature projections from any category, whereas the downstream task relies solely on projections

corresponding to the correct category. Note that, as shown in Fig. 2, introducing such transformations into surrogate model training can mitigate the domain shift, but incurs higher hardware costs and increased time consumption.

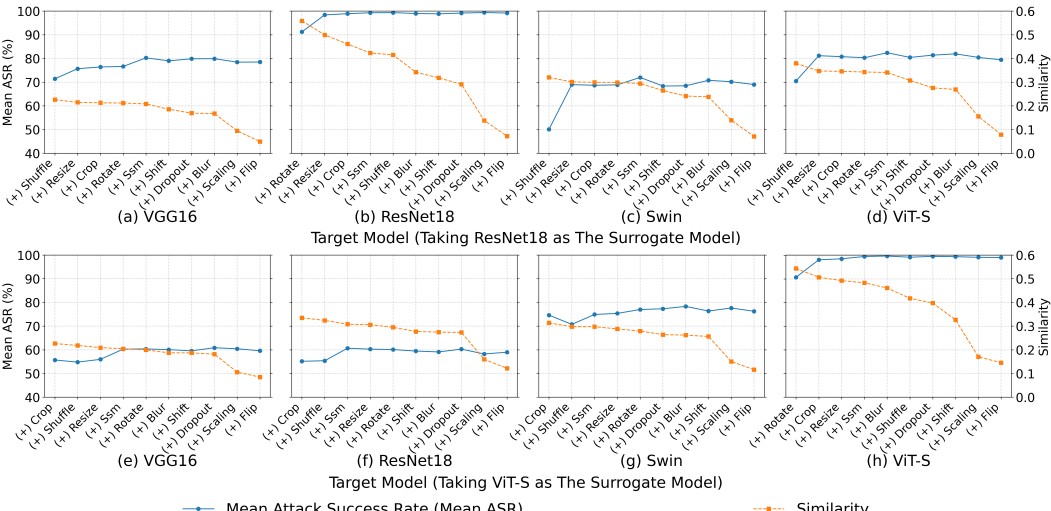

Figure 4: The similarity between adversarial examples generated through the approach in Eq. 5 and those produced using individual transformations. We then rank the transformations based on similarity and report the mean attack success rate for the ordered combination of these transformations. The similarity approximates each transformation's contribution. The transformations employed include block-shuffle Wang et al. (2024a), resize-padding Xie et al. (2019), random cropping, ssm Long et al. (2022) (which randomly scales images and adds noise in the frequency domain), random shift, dropout, Gaussian blurring, random scaling, and random flip. The models are trained on CIFAR-100 with rotation and resize-padding.

So far, we know that it is feasible to expand the function space through transformations that are not used in training. We next explore compensating for model differences through function space expansion centered on surrogate models. An ensemble attack-like task is introduced to generate adversarial examples, which can be formulated as

$$\frac{1}{I} \sum_i J(f_s(\varphi_i(x)), y) + J(f_{tar}(x), y), \tag{5}$$

where $f_{tar}$ denotes the target model, and the $\varphi_i$ represents a composite transformation sampled from a transformation set $\Psi$. We quantify the similarity between adversarial examples generated through the approach in Eq. 5 and those produced using individual transformations. The similarity approximates each transformation's contribution, with results visualized in Fig. 4. The results reveals that despite architectural differences among target models, transformation contribution rankings remain remarkably consistent, particularly for the least effective transformations. Although function space expansion via transformations exhibits some bias, transformations yielding limited function space expansion typically underperform across all target models.

### 3.3 PROPOSED INPUT TRANSFORMATION-BASED ATTACK

Based on our findings, we propose a simple transfer-based attack named SimAttack, as shown in Alg. 1. To maximize the function space, we randomly select several transformations from the transformation set $\Psi$ as a composite transformation $\varphi_i$ (Row 3 in Alg. 1), rather than serially combining all transformations or using other methods. Furthermore, we observe that domain shifts caused by different transformations may lead to gradient imbalances, as shown in Tab. 1. Specifically, gradients derived from transformations used during training tend to be smaller than those from other transformations. Therefore, inspired by optimizers Kingma & Ba (2014); Loshchilov & Hutter (2017), we introduce $l_2$-norm over each input as the normalization $Norm(\cdot)$ into our proposed attack (Row 4 in Alg. 1), thereby mitigating gradient imbalance.

Table 1: Gradients of inputs with different transformations. We average the results of 10,000 transformed data.

|  | Rotation | Resize-padding | Block Shuffle | Shift |
|---|---|---|---|---|
| ResNet18 | 0.0282 | 0.0207 | 0.0740 | 0.0830 |
| ViT-S | 0.0813 | 0.0793 | 0.1406 | 0.1427 |

As mentioned above, data transformation causes domain shifts, which in turn lead to increased noise in gradients. Increasing the number of random transformations mitigates noise, and we quantify this relationship in Fig. 5a and Fig. 5b. A single transformation may require more than 100 times to achieve optimal performance. Therefore, the transformation number may need to be set to several thousand to achieve optimal performance when using a number of transformations.

---

**Algorithm 1** A simple transfer-based attack (SimAttack)

---

**Input**: an benign example $x_0$, adversarial example $x_t^{adv}$, perturbation budget $\epsilon$, transformation set $\Psi$, step size $\alpha$, iteration $T$, transformation number $I$.
**Output**: adversarial example $x_T^{adv}$.

1: Initialize $\alpha = \epsilon/T$, $x_0^{adv} = x_0$, $\overline{g_0} = 0$.
2: **for** $t = 1$ **to** $T$ **do**
3:     Sample $I$ transformation compositions $\{\varphi_i\}_{i=0}^I$ from transformation set $\Psi$.
4:     Update $g_t = \sum\limits_{i}^{I} Norm(\nabla_{x_{t-1}^{adv}} J(f(\varphi_i(x_{t-1}^{adv})), y))$ .
5:     Update the momentum $\overline{g_t} = \overline{g_{t-1}} + \frac{g_t}{\|g_t\|_1}$.
6:     Update the example $x_t^{adv} = clip(x_{t-1}^{adv} + \alpha \cdot sign(\overline{g_t}), 0, 1)$.
7: **end for**
8: **return** the adversarial example $x_t^{adv}$

---

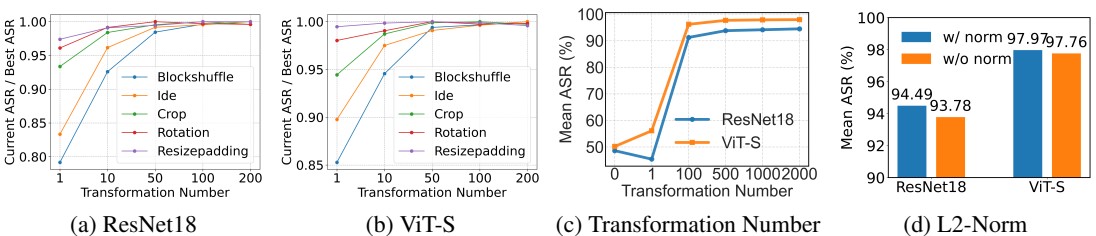

| (a) ResNet18 | (b) ViT-S | (c) Transformation Number | (d) L2-Norm |

Figure 5: **(a)** and **(b)**: The relationship between noise reduction and transformation number. The surrogate models are trained on CIFAR-100 with rotation and resize-padding. The term "Current ASR / Best ASR" refers to the current attack success rate divided by the best attack success rate. **(c)** and **(d)**: The role of transformation number $I$ and L2-norm.

## 4 EXPERIMENTS

### 4.1 EXPERIMENTAL SETUP

We describe the dataset, implementation setup, and input transformation setup in detail here.

**Implementation Setup.** Following the previous works Wang et al. (2021; 2023); Zhu et al. (2024a), we calculate attack success rate on 1, 000 images chosen from ILSVRC 2012 validation set Russakovsky et al. (2015), and these images are classified correctly by all the models. Following the widely used hyperparameter setup in the works Dong et al. (2018); Zhu et al. (2024a); Lin et al. (2024), we set the perturbation budget $\epsilon$ to 16/255, iteration number $T$ to 10, step size $\alpha$ to 1.6/255. By default, the transformation number $I$ is 2,000. We run the experiments on a single A100.

**Input Transformation Setup.** The properties of a model may primarily depend on its structure, so we follow the experience shown in Fig. 4. By default, the input transformation set $\Psi$ consists of block shuffle, resize-padding, SSM, random shift, random cropping, and random rotation. To expand the target function space, hyperparameters of input transformations are randomly selected as follows: Block-shuffle splits inputs into $1 \rightarrow 25$ randomly sized blocks; resize-padding adopts the configuration from Xie et al. (2019) with maximum scaling factor uniformly sampled from 1.14 to 1.66 (minimum fixed at 1.0); SSM applies noise factors sampled uniformly from 0.1 to 0.9; random shift magnitudes range from 0 to a predefined maximum; random cropping adds 0-30 pixels of padding before resizing to 224×224; and random rotation employs maximum angles uniformly distributed between 0 and 180.

## 4.2 ROLE OF HYPERPARAMETER AND L2-NORM

We show the role of the transformation number $I$ and L2-norm on our proposed SimAttack. The results are shown in Fig. 5. If the transformation number is too small, the gradient noise introduced by domain shifts may reduce performance.

## 4.3 CASE STUDY ON LLM

We take the ViT-B as the surrogate model and show the adversarial examples of SimAttack successfully attacking GPT 4.1 Achiam et al. (2023), as shown in Fig. 6. Large models still cannot escape the vulnerability of neural networks while people can still distinguish the images.

Figure 6: Case study on GPT 4.1 examples.

## 4.4 COMPARATIVE EXPERIMENTS

In this section, we adopt 5 common neural networks as surrogate models to compare our proposed AdaAES with other advanced attacks and evaluate the attack success rate of different transfer-based adversarial attacks on twelve models including ResNet18 He et al. (2016), ResNet50 He et al. (2016), ResNet101 He et al. (2016), ResNeXt50 Xie et al. (2017), DenseNet121 Huang et al. (2017), VGG19 Simonyan & Zisserman (2014), ViT-S Dosovitskiy et al. (2020), ViT-B Dosovitskiy et al. (2020), PiT-B Zhang et al. (2023), Visformer Chen et al. (2021), and Swin Transformer Liu et al. (2021). We pick 7 adversarial attacks (including MI-FGSM Dong et al. (2018), DEM Zou et al. (2020), SIA Wang et al. (2023), ANDA Fang et al. (2024), BSR Wang et al. (2024a), DeCowA Lin et al. (2024), L2T Zhu et al. (2024a), OPS Guo et al. (2025)) as the comparative methods. The results are shown in Tab. 2. The results show that our method outperforms all existing methods across multiple surrogate models, which supports the soundness of our work.

Table 2: Attack success rate (%) across twelve models on the adversarial examples crafted on a surrogate model (labeled "Sur.").

| Sur. | Attack | Res18 | Res50 | Res101 | NeXt | Den | VGG | Incv3 | ViT-S | ViT-B | PiT | Vis. | Swin | Mean |
|---|---|---|---|---|---|---|---|---|---|---|---|---|---|---|
| InceptionV3 | MI-FGSM | 47.3 | 30.0 | 28.1 | 28.5 | 44.5 | 47.9 | 97.9 | 23.1 | 13.7 | 16.9 | 24.3 | 28.8 | 35.9 |
| | DEM | 77.2 | 57.1 | 55.5 | 57.6 | 78.8 | 76.0 | 99.0 | 47.4 | 30.6 | 35.5 | 47.7 | 49.2 | 59.3 |
| | SIA | 87.9 | 69.2 | 65.4 | 69.0 | 85.9 | 83.6 | 99.9 | 49.1 | 34.7 | 46.5 | 58.9 | 61.5 | 67.6 |
| | ANDA | 66.1 | 50.1 | 48.4 | 49.8 | 69.5 | 66.0 | 99.7 | 38.1 | 27.2 | 31.8 | 42.9 | 45.6 | 52.9 |
| | BSR | 87.7 | 71.9 | 67.5 | 70.6 | 87.0 | 85.6 | 99.8 | 51.1 | 37.0 | 48.7 | 62.8 | 65.6 | 69.6 |
| | DeCowA | 78.7 | 57.8 | 57.3 | 61.1 | 78.5 | 78.8 | 98.0 | 47.4 | 32.1 | 38.9 | 49.6 | 54.7 | 61.1 |
| | L2T | 83.9 | 70.6 | 67.8 | 70.4 | 84.6 | 80.7 | 98.9 | 52.4 | 37.3 | 49.2 | 56.6 | 61.6 | 67.8 |
| | OPS | 96.2 | 83.6 | 85.1 | 85.5 | 95.9 | 91.9 | 99.9 | 77.3 | 59.9 | 65.4 | 77.9 | 80.7 | 83.3 |
| | Ours | **98.2** | **92.3** | **91.8** | **93.3** | **98.1** | **98.0** | **100.0** | **82.9** | **71.0** | **77.3** | **87.3** | **87.8** | **89.8** |
| DenseNet121 | MI-FGSM | 74.9 | 61.5 | 50.9 | 55.2 | 99.9 | 68.5 | 58.0 | 31.6 | 20.6 | 27.9 | 41.4 | 44.3 | 52.9 |
| | DEM | 98.0 | 91.0 | 85.8 | 89.1 | 99.9 | 94.4 | 94.2 | 63.7 | 48.8 | 52.8 | 75.4 | 70.2 | 80.3 |
| | SIA | 98.6 | 95.6 | 92.2 | 94.9 | **100.0** | 97.6 | 91.9 | 64.6 | 48.3 | 67.5 | 84.6 | 81.7 | 84.8 |
| | ANDA | 93.4 | 86.2 | 81.0 | 83.6 | 99.9 | 89.8 | 82.6 | 53.7 | 40.8 | 55.3 | 71.0 | 69.8 | 75.6 |
| | BSR | 98.6 | 95.0 | 89.6 | 93.1 | **100.0** | 97.1 | 88.2 | 62.6 | 49.1 | 66.3 | 83.5 | 79.8 | 83.6 |
| | DeCowA | 98.5 | 92.5 | 89.0 | 91.4 | **100.0** | 96.4 | 93.8 | 73.3 | 57.7 | 70.3 | 83.4 | 80.6 | 85.6 |
| | L2T | 98.8 | 95.0 | 92.9 | 94.2 | **100.0** | 97.7 | 94.4 | 74.6 | 59.1 | 73.3 | 85.6 | 85.7 | 87.6 |
| | OPS | 99.9 | 98.2 | 98.5 | 98.3 | **100.0** | 98.7 | 99.2 | 90.6 | 79.8 | 86.0 | 94.7 | 93.4 | 94.8 |
| | Ours | **100.0** | **99.9** | **99.7** | **99.7** | **100.0** | **99.8** | **99.7** | **94.0** | **85.2** | **91.0** | **97.4** | **96.6** | **96.9** |
| ResNet18 | MI-FGSM | **100.0** | 49.3 | 42.2 | 45.7 | 73.8 | 74.4 | 55.6 | 27.6 | 16.7 | 23.0 | 32.6 | 40.1 | 48.4 |
| | DEM | **100.0** | 82.5 | 76.8 | 81.8 | 97.5 | 95.1 | 92.1 | 58.7 | 39.1 | 46.0 | 66.3 | 65.9 | 75.2 |
| | SIA | **100.0** | 91.9 | 87.6 | 89.7 | 99.2 | 98.6 | 91.5 | 62.7 | 43.9 | 58.5 | 77.3 | 77.0 | 81.5 |
| | ANDA | **100.0** | 80.5 | 74.7 | 78.6 | 96.6 | 94.8 | 85.6 | 53.1 | 38.6 | 49.5 | 66.1 | 68.8 | 73.9 |
| | BSR | **100.0** | 90.5 | 86.0 | 88.4 | 98.8 | 98.7 | 90.3 | 60.8 | 43.0 | 57.9 | 77.3 | 75.9 | 80.6 |
| | DeCowA | **100.0** | 89.0 | 85.0 | 88.3 | 98.5 | 98.4 | 94.4 | 72.3 | 56.5 | 63.7 | 80.5 | 79.8 | 83.9 |
| | L2T | **100.0** | 91.5 | 87.6 | 91.6 | 98.6 | 98.8 | 94.8 | 67.4 | 51.0 | 64.7 | 78.8 | 81.2 | 83.8 |
| | OPS | **100.0** | 97.2 | 96.9 | 97.1 | 99.9 | 99.6 | 99.0 | **91.4** | **77.1** | 81.5 | 93.0 | 91.7 | 93.7 |
| | Ours | **100.0** | **98.6** | **98.1** | **98.7** | **100.0** | **99.9** | **99.2** | 90.6 | 76.5 | **83.9** | **94.8** | **93.6** | **94.5** |
| ViT-S | MI-FGSM | 51.4 | 33.6 | 30.3 | 33.8 | 48.9 | 54.7 | 45.0 | **100.0** | 69.2 | 37.4 | 42.6 | 54.1 | 50.1 |
| | DEM | 88.8 | 81.4 | 79.7 | 81.9 | 89.2 | 88.0 | 90.3 | 99.9 | 95.2 | 88.1 | 88.1 | 90.4 | 88.4 |
| | SIA | 86.2 | 80.3 | 76.4 | 78.3 | 87.4 | 85.8 | 80.6 | **100.0** | 95.7 | 84.9 | 86.0 | 90.3 | 86.0 |
| | ANDA | 70.7 | 60.8 | 57.4 | 60.8 | 73.3 | 71.0 | 67.4 | **100.0** | 89.1 | 67.5 | 69.7 | 77.1 | 72.1 |
| | BSR | 87.6 | 82.4 | 82.0 | 83.6 | 89.0 | 87.1 | 84.0 | **100.0** | 94.8 | 90.6 | 88.1 | 91.1 | 88.4 |
| | DeCowA | 86.0 | 75.7 | 73.8 | 77.5 | 97.1 | 85.3 | 84.2 | 98.8 | 87.2 | 83.4 | 83.6 | 85.9 | 84.9 |
| | L2T | 88.5 | 81.1 | 78.0 | 80.8 | 88.0 | 87.1 | 86.7 | 99.2 | 92.8 | 84.5 | 84.5 | 89.6 | 86.7 |
| | OPS | 96.1 | 91.1 | 93.0 | 92.8 | 96.6 | 94.2 | 97.2 | 99.7 | 97.0 | 95.0 | 95.2 | 96.0 | 95.3 |
| | Ours | **98.1** | **95.7** | **96.0** | **97.3** | **98.5** | **97.6** | **98.3** | **100.0** | **98.9** | **98.1** | **98.3** | **98.8** | **98.0** |
| ViT-B | MI-FGSM | 52.8 | 39.3 | 33.8 | 38.8 | 50.9 | 57.3 | 46.4 | 72.0 | 97.3 | 40.5 | 43.4 | 54.7 | 52.3 |
| | DEM | 85.1 | 77.8 | 78.5 | 78.4 | 87.4 | 85.3 | 86.3 | 93.7 | 97.9 | 86.9 | 85.2 | 85.9 | 85.7 |
| | SIA | 77.4 | 75.2 | 72.8 | 76.1 | 80.5 | 79.0 | 76.0 | 90.4 | 97.3 | 81.4 | 81.4 | 84.5 | 81.0 |
| | ANDA | 67.0 | 60.1 | 58.9 | 60.9 | 70.9 | 69.1 | 66.4 | 84.3 | 97.7 | 66.7 | 68.0 | 73.1 | 70.3 |
| | BSR | 74.9 | 73.7 | 71.7 | 73.2 | 78.4 | 75.2 | 75.3 | 84.1 | 93.9 | 78.2 | 76.0 | 79.7 | 77.9 |
| | DeCowA | 82.1 | 74.3 | 74.1 | 76.0 | 81.8 | 79.1 | 81.4 | 86.7 | 92.2 | 83.1 | 82.4 | 82.6 | 81.3 |
| | L2T | 82.9 | 78.2 | 76.7 | 77.9 | 83.0 | 82.3 | 82.0 | 90.2 | 95.7 | 82.2 | 82.6 | 85.5 | 83.3 |
| | OPS | 94.3 | 91.8 | 91.8 | 92.8 | 95.5 | 92.2 | 94.4 | 98.0 | 98.7 | 95.0 | 94.9 | 95.1 | 94.5 |
| | Ours | **97.7** | **96.4** | **96.8** | **97.3** | **98.4** | **97.4** | **97.8** | **98.9** | **99.7** | **98.4** | **98.2** | **98.2** | **97.9** |

## 5 CONCLUSIONS

This work reveals that: 1) To achieve model transformation-invariance, data transformations establish multiple projections from transformed inputs to outputs, rather than enabling models to capture invariant features during training. This suggests introducing the transformations into surrogate models. 2) Input transformation-based attacks leverage transformations to expand target function space, thereby improving adversarial transferability. The transform's ability to expand the function space may provide guidance across datasets. Also, normalization should be incorporated into the paradigm of the attacks due to gradient imbalance.

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

# A APPENDIX

## A.1 ABLATION STUDY ON INPUT TRANSFORMATION

By default, the input transformation set $\Psi$ employed by our proposed SimAttack consists of block shuffle, resize-padding, SSM, random shift, random cropping, and random rotation. We further demonstrate the impact of adding and removing input transformations on our proposed SimAttack, as shown in Tabs. 3 and 4. The results supports that the selected transformation is the best setup.

Table 3: Attack success rate (%) across twelve models on the adversarial examples crafted on ResNet18 by our proposed SimAttack with different transformations. By default, the input transformation set $\Psi$ employed by our proposed SimAttack consists of block shuffle, resize-padding, SSM, random shift, random cropping, and random rotation.

| Method | Res18 | Res50 | Res101 | NeXt | Den | VGG | Incv3 | ViT-S | ViT-B | PiT | Vis. | Swin | Mean |
|---|---|---|---|---|---|---|---|---|---|---|---|---|---|
| - Random shift | 100 | 97.8 | 96.7 | 97.6 | 100 | 99.9 | 98.9 | 89.2 | 73.0 | 81.9 | 93.1 | 92.4 | 93.4 |
| SimAttack (Ours) | 100 | 98.6 | 98.1 | 98.7 | 100 | 99.9 | 99.2 | 90.6 | 76.5 | 83.9 | 94.8 | 93.6 | 94.5 |
| + Gaussian blurring | 100 | 97.4 | 96.6 | 97.5 | 100 | 99.8 | 98.5 | 89.2 | 73.3 | 81.3 | 92.6 | 90.9 | 93.1 |
| + Dropout | 100 | 97.0 | 95.5 | 96.8 | 99.9 | 99.9 | 98.5 | 88.5 | 71.2 | 79.8 | 92.7 | 90.7 | 92.5 |
| + Random scaling | 100 | 96.5 | 95.7 | 96.6 | 99.9 | 99.4 | 98.5 | 87.5 | 70.8 | 78.6 | 92.1 | 90.2 | 92.2 |
| + Random Flip | 100 | 97.4 | 96.7 | 97.4 | 99.9 | 99.5 | 98.8 | 89.2 | 73.2 | 80.3 | 92.3 | 91.6 | 93.0 |

Table 4: Attack success rate (%) across twelve models on the adversarial examples crafted on ViT-S by our proposed SimAttack with different transformations. By default, the input transformation set $\Psi$ employed by our proposed SimAttack consists of block shuffle, resize-padding, SSM, random shift, random cropping, and random rotation.

| Method | Res18 | Res50 | Res101 | NeXt | Den | VGG | Incv3 | ViT-S | ViT-B | PiT | Vis. | Swin | Mean |
|---|---|---|---|---|---|---|---|---|---|---|---|---|---|
| - Random shift | 98.2 | 95.3 | 96.2 | 96.7 | 98.3 | 97.6 | 98.5 | 100 | 98.8 | 97.8 | 98.1 | 99.0 | 97.9 |
| SimAttack (Ours) | 98.1 | 95.9 | 96.1 | 97.2 | 98.5 | 97.6 | 98.3 | 100 | 98.9 | 98.1 | 98.4 | 98.8 | 98.0 |
| + Gaussian blurring | 97.6 | 93.3 | 93.5 | 95.3 | 98.5 | 96.1 | 97.4 | 100 | 98.5 | 97.6 | 96.5 | 98.1 | 96.9 |
| + Dropout | 97.2 | 93.0 | 92.8 | 94.5 | 98.1 | 96.0 | 97.0 | 100 | 98.6 | 97.1 | 96.3 | 98.0 | 96.6 |
| + Random scaling | 97.2 | 91.5 | 92.1 | 93.1 | 97.5 | 95.6 | 96.3 | 100 | 98.3 | 96.7 | 95.5 | 97.2 | 95.9 |
| + Random Flip | 97.3 | 92.0 | 91.8 | 93.6 | 97.3 | 95.8 | 96.8 | 100 | 98.4 | 96.5 | 95.7 | 97.1 | 96.0 |

## A.2 CASE STUDY ON LLM

We take 5 neural networks (i.e., DenseNet18, InceptionV3, ResNet18, ViT-S and ViT-B) as the surrogate models, respectively, and show the adversarial examples of SimAttack successfully attacking GPT 4.1, as shown in Figs. 7 and 8. The surrogate models are trained on ImageNet dataset. Overall, large models still cannot escape the vulnerability of neural networks while people can still distinguish the images. In Addition, the results in Row 1 (right) suggests that the two models based on the transform structure may be more effective than convolutional neural networks (CNNs) in attacking large language models, since the adversarial examples generated by ViT-S and ViT-B successfully attack the GPT 4.1 but the ones generated by DenseNet121, InceptionV3 and ResNet18 are unable to do so.

## A.3 THE USE OF LARGE LANGUAGE MODELS

We employ large language models to check for typos and grammatical errors, and utilize the LLMs to provide suggestions for fixing code bugs.

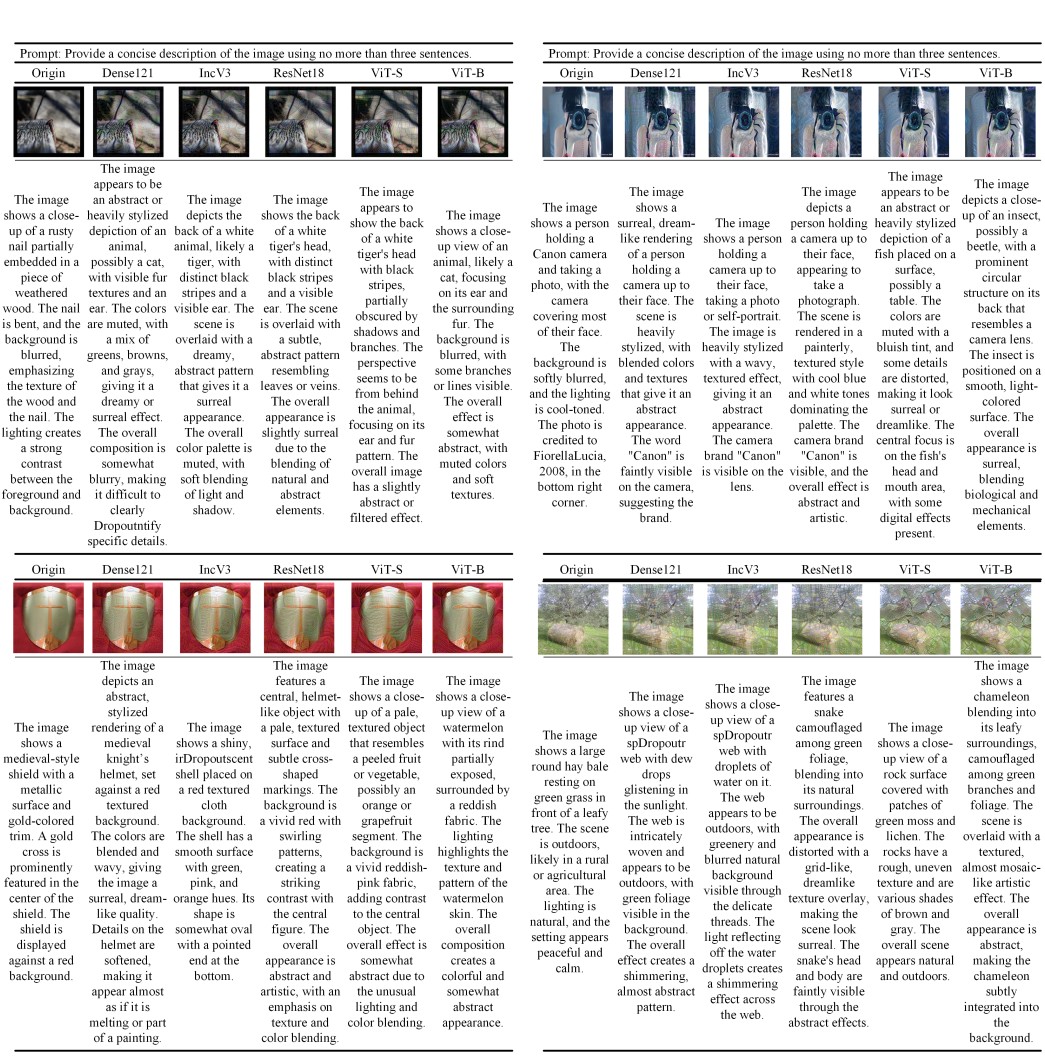

Figure 7: Case study on GPT 4.1 examples.

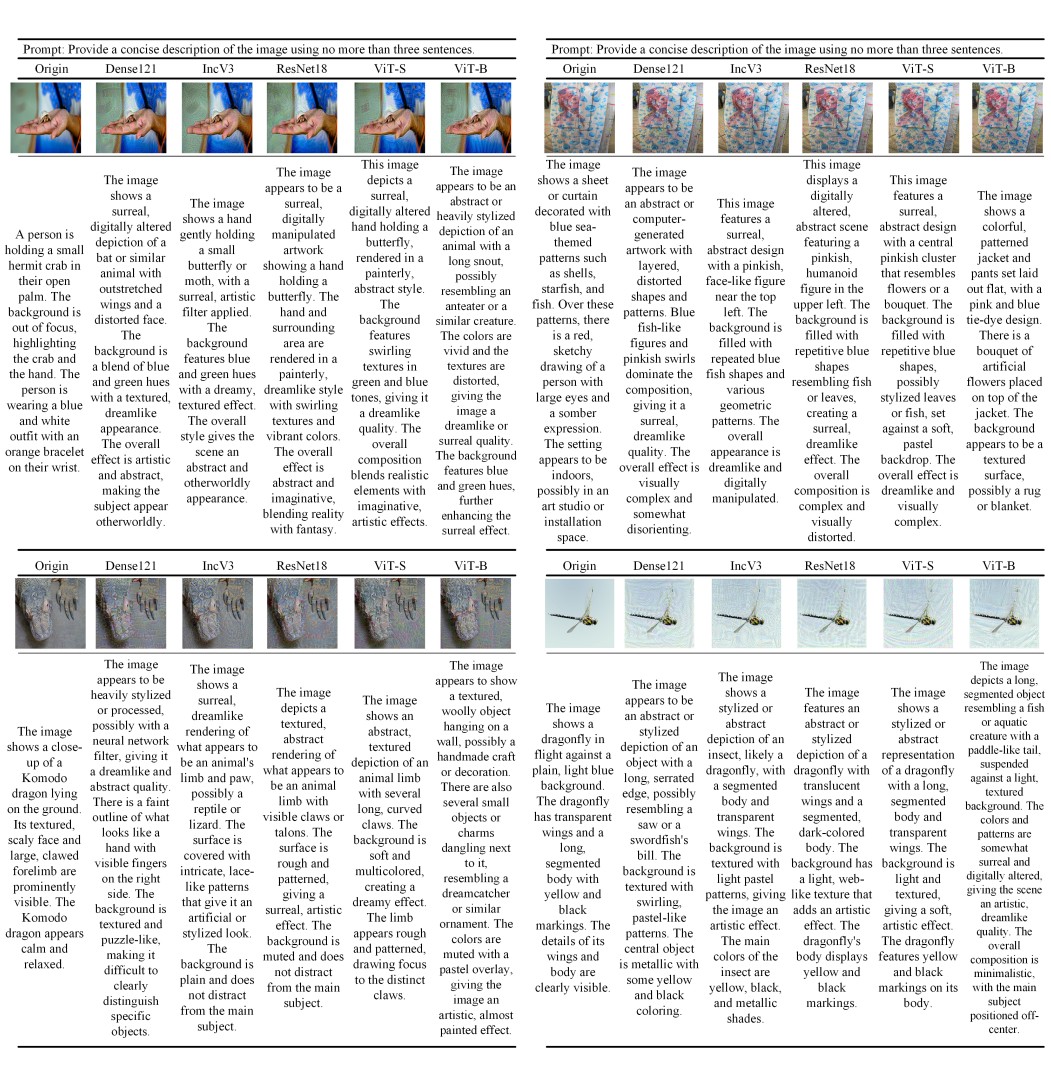

Figure 8: Case study on GPT 4.1 examples.