# OpenReview forum: "Understanding Input Transformation-Based Attacks via Target Function Space Expansion"
_ICLR.cc/2026/Conference — ICLR 2026 Conference Withdrawn Submission_

### Official Review · Reviewer_j7Ax · 2025-10-26

**Soundness:** 2
**Presentation:** 3
**Contribution:** 2
**Rating:** 4
**Confidence:** 4

**Summary:**

This paper investigates the mechanism behind input transformation-based adversarial attacks, arguing that such methods enhance transferability by expanding the target function space. The authors also propose the use of L2 normalization to address gradient imbalance during attack generation. Based on these insights, they introduce SimAttack, a simple yet effective method that achieves state-of-the-art performance.

**Strengths:**

1、The paper is well-organized, clearly written, and accessible. The logical flow from motivation to methodology and experiments is smooth and easy to follow.

2、A key finding is that surrogate models trained with input transformations produce adversarial examples with significantly higher transferability than those trained without. This insight is valuable and empirically supported.

3、The proposed SimAttack demonstrates strong performance across multiple surrogate and target models, outperforming existing methods in most settings.

**Weaknesses:**

1、The authors conclude that input transformations improve transferability by increasing feature diversity rather than extracting transformation-invariant features. This claim seems overly absolute. An alternative interpretation is that both mechanisms—diversity and invariance—may work together. For instance, Figures 1 and 4 could support the idea that increasing transformations help the model focus on more robust and invariant features by filtering out noise or irrelevant variations.

2、The concept of "function space expansion" is not fully grounded. A more formal or intuitive explanation of how transformations expand the function space and why this improves transferability would strengthen the theoretical contribution.

3、The claim that models do not learn transformation-invariant features is strong but not thoroughly verified. More evidence across different model architectures and transformation types would make the argument more convincing.

**Questions:**

1、Consider a more balanced interpretation of how input transformations work, acknowledging the potential role of both feature diversity and invariance.

2、Provide a deeper theoretical or empirical analysis of how function space expansion improves transferability.

3、Extend the validation of the "no invariant features" claim with additional experiments or references.

---

### Official Review · Reviewer_ZZS9 · 2025-10-29

**Soundness:** 2
**Presentation:** 1
**Contribution:** 2
**Rating:** 2
**Confidence:** 5

**Summary:**

This work aims to understand the previous poorly-understood aspects input-transformation-based adversarial attacks. The authors proposed SimAttack that achieves an average attack access rate of 95.4% on both CNN- and ViT-based models.

**Strengths:**

- The paper addresses an important yet under explored aspect in input transformation-based attacks.
- Interesting idea for adversarial attacks.

**Weaknesses:**

Thank you for the great efforts to the paper. However, the review has to give low ratings due to the following reasons:

1. The paper suffers from poor writing, and it is hard to follow due to English wording choices. At least major re-writing is required before it is ready for publication. The review lists some of the issues below.
    - In contribution statement (section 1). Point 2 is repetitive and should be merged with or revised to distinguish it from Point 1.
    - "Please correct the non-standard citation format in Section 2. The current in-text citations, such as 'SGM Wu et al. (2020)' and 'DIM Xie et al.', do not follow any established guidelines."
    - What does “shuffle (3)” mean in Lines 179-180? The parameter (3) is left unexplained.
    - In lines 265-266, what does “enhancing the bound of random noise” mean? Raising the magnitude?
    - In lines 301-302, what does “compensating for model differences” mean?
    - In lines 311-312, what does “bias” mean?
    - The wording choice “Transformation number $I$” in lines 254-255 is weird. It should be “transformation count”.
    - How  “noise reduction” is related to Figure 5?
2. Method soundness.
    - The paper proposes to understand transformation in a new function space. A new prospective is that “transformation modifies the surrogate model, not the input data” is very interesting. However, it is not clear that how this point is illustrated in further sections, e.g. Section 3.2. How is figure 4 related to “function space expansion”?
    - The attack formulation in SimAttack (Alg. 1) is wrong. Step 4 describes $g_t$ as a scalar (since a norm is a scalar) instead of a vector. The dimension does not match. Hence the total algorithm is barely understandable.
3. Evaluation
    - As a tradition of adversarial attack literature, it would be better to include a combination of adversarial attacks for better comparison, say SimAttack + PIM + DIM.
    - The evaluation on multi-modal models is limited. Only a case study (without elaboration on experimental setup) is discussed.

**Questions:**

Please respond to the concerns in "Weakness" section. The reviewer will raise my rating if my concern gets solved.

---

### Official Review · Reviewer_1NMG · 2025-10-31

**Soundness:** 2
**Presentation:** 2
**Contribution:** 2
**Rating:** 2
**Confidence:** 4

**Summary:**

This paper explores why input-transformation-based transferable attacks (e.g., DIM, SIM, TIM) are effective. The authors argue that transformations expand the *target function space* rather than enforcing transformation invariance. They propose **SimAttack**, which applies multiple random transformations per iteration, normalizes each gradient by its L2 norm, and accumulates momentum. Experiments on ImageNet show improved transferability across 12 models compared with prior methods.

**Strengths:**

- Provides an intuitive conceptual framing of transformation-based attacks as *function-space expansion*, potentially unifying several heuristic approaches.
- SimAttack is simple to implement and empirically stable across multiple benchmarks.
- Broad comparisons are conducted against several known baselines.

**Weaknesses:**

1. **Unfair and uninformative experimental setup**
   - The paper uses **I = 2000 transformations per iteration**, whereas most comparable works (e.g., *Learning to Transform Dynamically for Better Adversarial Transferability*, *Improving the Transferability of Adversarial Samples by Path-Augmented  Method*) use **≤10 transformations** with equal-length augmentation chains for fairness.
   - Unlike prior works that optimize transformation sequences, SimAttack introduces **no selection strategy or adaptive mechanism**—it simply stacks thousands of random transforms.

2. **Questionable mechanistic evidence (Fig. 1 and Fig. 3)**
   - **Fig. 1**: Claims that models “produce invariant outputs but rely on different features,” yet gradient similarity is computed **without spatial realignment (\(\phi^{-1}\))**, so low cosine similarity could simply reflect pixel misalignment.
   - **Fig. 3**: Uses **uniform noise as a surrogate transformation**, which is unrealistic and does not represent genuine geometric or photometric domain shifts.
   - Since these two figures underpin the claimed “function-space expansion” mechanism, their methodological flaws **undermine the core argument**.
   - The “function-space expansion” idea is intuitive but lacks formal definition or measurable evidence (e.g., gradient diversity, representation distance).

3. **Limited novelty**
   - Applying multiple augmentations per iteration is already standard practice in input-diversification attacks.
   - The proposed method adds no new learning component, optimization strategy, or theoretical formulation.
   - Without formal analysis of “function-space expansion,” the contribution remains descriptive and incremental.

4. **Language and clarity issues**
   Overall writing quality requires substantial polishing.
   - **L33–L35:** “*models can improve transformation invariance by capturing diverse features … rather than transformation-invariant features*” — need clarification.
   - **L274:** Incorrect capitalization (*In Addition*).
   - **L420 / L546:** Inconsistent notation (*L2-norm* vs *l2-norm*).

**Questions:**

1. In **Fig. 1**, were gradients spatially realigned (via \(\phi^{-1}\)) before cosine similarity? Without alignment, low similarity may result from pixel shifts, not feature changes.
2. In **Fig. 3**, why use *uniform noise* as a transformation? Would geometric or photometric transforms (e.g., rotation, blur) better support the “expansion–domain-shift” claim?
3. SimAttack uses **I = 2000** transformations per iteration, while others use ≤10. Could you report results under equal computational budgets or chain lengths?
4. Is there any quantitative metric (e.g., gradient diversity or feature-space distance) to substantiate the “function-space expansion” hypothesis?
5. How sensitive is transferability to \(I\)? Does performance drop sharply when \(I\) is reduced?

---

### Official Review · Reviewer_H8pD · 2025-11-01

**Soundness:** 2
**Presentation:** 2
**Contribution:** 2
**Rating:** 4
**Confidence:** 4

**Summary:**

This paper investigates the underlying mechanisms through which input transformation-based adversarial attacks improve attack transferability. Such attacks enhance the transferability of adversarial examples by incorporating input transformations during the generation process. Through experimental analysis, the authors demonstrate that the improved generalization of models trained with input transformations does not stem from the prevailing view that “input transformations help models learn transformation-invariant features.” Instead, they argue that models learn more diverse feature representations from transformed inputs, thereby effectively expanding the mapping between input and output spaces. Consequently, applying input transformation-based attacks to surrogate models can be understood as expanding the target function space of the attack, which in turn enhances the generalization of adversarial examples. Building on this insight, the paper proposes a simple transformation-based attack method called SimAttack. The method randomly samples a large number of composite transformations, aggregates the L2-normalized gradients computed on each transformed input, and incorporates momentum updates. The L2 normalization and momentum updates help mitigate domain shifts introduced by different transformations. Experimental results show that SimAttack achieves state-of-the-art (SOTA) attack performance compared to existing input transformation-based attacks and is capable of transferring successfully even to GPT-4.1.

**Strengths:**

1.	This paper reinterprets the principle of input transformation-based attacks from the perspective of function space expansion. It further identifies that domain shifts introduced by different transformations cause gradient imbalance, which negatively affects the transferability of adversarial examples. To address this issue, the paper proposes using L2-normalized gradients as a solution.
2.	In the theoretical analysis section, the paper supports its arguments through well-designed experiments and insightful analysis. In the experimental section, it validates the effectiveness of the proposed method across various datasets and model architectures.
3.	The work provides a novel interpretive perspective for future studies on input transformation-based adversarial attacks.

**Weaknesses:**

1.	As noted in the Related Work, existing methods such as L2T, BSR, and OPS already apply input transformations during adversarial example generation to improve transferability. The proposed SimAttack shows no substantial difference from these methods — its only distinguishing feature appears to be the use of conventional L2-norm — which weakens its novelty. The authors should provide a detailed explanation of why SimAttack is effective and clearly articulate how it differs from prior work to strengthen the paper’s originality and persuasive power.
2.	The authors rely only on gradients and cosine similarity to argue that models do not learn input-invariant features; the paper lacks an examination of internal and output feature representations. The authors should further demonstrate whether the models actually learn input invariance.
3.	For the experiments in Figure 2, applying data augmentation during adversarial-example generation on a model that was not trained with such augmentation improves attack performance. This seems to contradict the statement on line 66: “If the surrogate model fails to extract meaningful features from these transformed data φ_i(x^adv_{t−1}), the resulting updates may become severely noisy.” In other words, augmented inputs that the surrogate model has never seen should hinder generation of meaningful perturbations, yet the results show improved performance. Please reconcile or clarify this apparent inconsistency.
4.	The computational cost is high: the default number of transformations I = 2000 leads to a very large computational burden, which would limit applicability in low-latency or high-throughput scenarios. The paper lacks experiments comparing compute resources and wall-clock time against baselines to substantiate claims about the method’s efficiency.
5.	The paper lacks ablation studies to demonstrate the effect of operations such as L2-normalization on improving adversarial transferability. Including such experiments would help clarify the contribution of each component to the overall performance.
6.	The paper does not evaluate performance against more robust defenses, for example adversarially trained models. Tests on stronger defenses are needed to assess the method’s effectiveness under more realistic, robust settings.
7.	The writing is not sufficiently clear in places, which impedes understanding. In particular, frequent switches between describing model training and adversarial example generation phases cause confusion; please improve clarity and explicitly distinguish these stages throughout the manuscript.
8.	There are some typographical errors — for example, Section 4.4 refers to “our proposed AdaAES” instead of the method presented in this paper. Please proofread and correct such mistakes.

**Questions:**

The specific suggestions are as outlined in the Weaknesses section above. Here, I list the issues I am particularly concerned about:

1.	I am curious why SimAttack is so effective, and what innovations it offers compared to existing methods.

2.	From the Gradient maps, although the model’s gradients change after adding data augmentation, it is still possible to observe that the model can extract core information, such as the outline of a sunflower. Does this imply that the model is still capturing invariant features?

3.	As mentioned in Weaknesses 2, I feel somewhat confused about the boundary between noisy and meaningful signals.

4.	I would like to know the differences between this work and existing methods in terms of time and memory (GPU) requirements.

---

### Note · Authors · 2025-11-19

I have read and agree with the venue's withdrawal policy on behalf of myself and my co-authors.